# Unveiling the Role of RNA Recognition Motif Proteins in Orchestrating Nucleotide-Binding Site and Leucine-Rich Repeat Protein Gene Pairs and Chloroplast Immunity Pathways: Insights into Plant Defense Mechanisms

**DOI:** 10.3390/ijms25105557

**Published:** 2024-05-20

**Authors:** Fengwei Gu, Zhikai Han, Xiaodi Zou, Huabin Xie, Chun Chen, Cuihong Huang, Tao Guo, Jiafeng Wang, Hui Wang

**Affiliations:** 1College of Agriculture, South China Agricultural University, Guangzhou 510642, China; 99g999999ufw@gmail.com (F.G.); zhikai@stu.scau.edu.cn (Z.H.); 20232015065@stu.scau.edu.cn (X.Z.); 20221015020@stu.scau.edu.cn (H.X.); chchun@scau.edu.cn (C.C.); hchong@scau.cn (C.H.); guoguot@scau.edu.cn (T.G.); 2Nation Engineering Research Center of Plant Space Breeding, South China Agricultural University, Guangzhou 510642, China

**Keywords:** rice blast, NLRs, RNA recognition motif, chloroplast immunity

## Abstract

In plants, nucleotide-binding site and leucine-rich repeat proteins (NLRs) play pivotal roles in effector-triggered immunity (ETI). However, the precise mechanisms underlying NLR-mediated disease resistance remain elusive. Previous studies have demonstrated that the NLR gene pair *Pik-H4* confers resistance to rice blast disease by interacting with the transcription factor OsBIHD1, consequently leading to the upregulation of hormone pathways. In the present study, we identified an RNA recognition motif (RRM) protein, OsRRM2, which interacted with Pik_1_-H4 and Pik_2_-H4 in vesicles and chloroplasts. OsRRM2 exhibited a modest influence on *Pik-H4*-mediated rice blast resistance by upregulating resistance genes and genes associated with chloroplast immunity. Moreover, the RNA-binding sequence of OsRRM2 was elucidated using systematic evolution of ligands by exponential enrichment. Transcriptome analysis further indicated that OsRRM2 promoted RNA editing of the chloroplastic gene *ndhB*. Collectively, our findings uncovered a chloroplastic RRM protein that facilitated the translocation of the NLR gene pair and modulated chloroplast immunity, thereby bridging the gap between ETI and chloroplast immunity.

## 1. Introduction

Plants have evolved intricate defense mechanisms to detect and counteract invading pathogens, including viruses, bacteria, fungi, and pests. Plant immunity comprises two primary systems: pathogen-associated molecular pattern (PAMP)-triggered immunity (PTI) and effector-triggered immunity (ETI). PTI acts as the initial line of defense, activated upon recognition of conserved PAMPs by plant pattern recognition receptors (PRRs) [1,2]. Conversely, ETI represents a more targeted and robust defense mechanism triggered by the recognition of pathogen-specific effectors by intracellular receptors known as resistance (R) proteins [3,4]. These R proteins, predominantly members of the nucleotide-binding site and leucine-rich repeat (NLR) family, are categorized into two subfamilies, Toll-like/interleukin-1 receptor (TIR)-type NLRs (TNLs) and coiled-coil (CC)-type NLRs (CNLs), based on their domain composition [5].

Certain TNLs serve as NAD^+^-hydrolyzing enzymes, participate in signaling, and facilitate cooperation with helper NLRs [6]. Notably, CNLs such as ZAR1 [7,8] and Sr35 [9,10] have been shown to form resistosomes localized in the plasma membrane, characterized as Ca^2+^ channels. Moreover, NLRs contribute to transcriptional reprogramming through interactions with transcription factors (TFs) [11,12,13] and can translocate to vesicles. For instance, upon infection with AvrPib, the interaction between the CNL Pib and the SH3 protein SH3P2 in the vesicles is attenuated, resulting in the release of Pib into the cytoplasm and triggering a hypersensitive response (HR) [14]. Additionally, the maize ESCRT protein ZmVPS23L facilitates the translocation of the CNL Rp1-D21 from the cytoplasm to the vesicles, thereby suppressing HR signaling in the absence of infection [15].

Recent investigations have unveiled the interactions between ETI and chloroplasts. For example, potato CNL *Rpi-vnt1.1*-mediated resistance is dependent on light and requires a chloroplast-located protein, GLYK [16]. Similarly, the CNL N’ from *Nicotiana benthamiana* promotes the degradation of a chloroplast-located protein THF1, a negative regulator of HR, through direct interaction [17]. While many NLRs are predominantly located in the cytoplasm and nucleus and interact directly with TFs or resistance-related components, reports on the translocation and chloroplast localization of NLRs remain scarce.

Rice blast, caused by *Magnaporthe oryzae*, reduces rice production by approximately 10% to 30% annually [18], posing a significant threat to global food security. The most effective and environmentally sustainable approach to mitigating losses from rice blast disease is breeding rice cultivars carrying broad-spectrum resistance genes against rice blast [19].

One such resistance gene pair, *PigmR*/*PigmS*, belonging to the CNL family, confers broad-spectrum rice blast resistance [20,21] by interacting with an RNA recognition motif (RRM) protein, PIBP1 [13]. In *Arabidopsis*, the RRM proteins GRP7 [22] and CPR5 [23] are also involved in chloroplast immunity challenge with *Pseudomonas syringae*. In previous studies, we have identified and fine-mapped a synonymous mutation, *Pik-H4*, from a mutant, as well as a CNL rice blast broad-spectrum resistance gene pair [24]. The *Pik* locus comprises a pair of CNL genes arranged in a head-to-head manner, collectively conferring *Pik*-mediated resistance [25,26,27]. Pik_1_-H4 has been shown to induce transcriptional reprogramming by interacting with a homeodomain TF, OsBIHD1, resulting in the upregulation of genes involved in ethylene and brassinosteroid synthesis [28]. However, the precise functional mechanism of the *Pik-H4* gene pair remains elusive.

In this study, we identified the interaction of a rice RRM protein encoded by LOC_Os03g25960 [referred to as *Oryza sativa* RNA recognition motif 2 (OsRRM2) in this study] with both Pik_1_-H4 and Pik_2_-H4, with observations of this interaction occurring within the vesicles and chloroplasts. Furthermore, we demonstrated the upregulation of pathways related to chloroplast immunity in transgenic plants overexpressing *OsRRM2*. Additionally, we identified the binding sites of OsRRM2 and RNA-editing events within the chloroplast proteins. These findings collectively provide insight into the involvement of RRM proteins in NLR-mediated chloroplast immunity.

## 2. Results

### 2.1. Pik_1_-H4 and Pik_2_-H4 Interacts with OsRRM2

In our previous research aimed at elucidating the rice blast resistance mechanism of Pik-H4 [28], we conducted a screening of a rice cDNA library using the yeast two-hybrid (Y2H) system. Through this approach, we identified an interaction between an RRM family protein encoded by LOC_Os03g25960 (henceforth referred to as OsRRM2) and both Pik_1_-H4 and Pik_2_-H4 (Figure 1a). Given that the coiled-coil (CC) domains of CNLs have been reported to interact with multiple proteins [13,28,29], we further investigated whether the CC domain of *Pik-H4* was involved in the interaction with OsRRM2. To this end, we split Pik_1_-H4 and Pik_2_-H4 and assessed their interaction with OsRRM2-CC in Y2H assays. Moreover, the OsRRM2-glutathione S-transferase (GST) fusion protein and the Pik_1_-H4-CC-His or Pik_2_-H4-CC-His fusion proteins were generated in *Escherichia coli* BL21. The interaction between Pik_1_-H4 and Pik_2_-H4 and OsRRM2 was subsequently confirmed using in vitro pulldown assays (Figure 1b). As OsRRM2 belongs to the RRM family, it is plausible that specific domains within OsRRM2 are responsible for interacting with the CC domains of Pik_1_-H4 and Pik_2_-H4. Additional Y2H experiments involving split OsRRM2, guided by Pfam database analysis (Appendix A), revealed that amino acids 1-85 (weak interaction) and 86-179 of OsRRM2 interacted with the CC domains of Pik_1_-H4 and Pik_2_-H4 in yeast (Figure 1a, Appendix A). These results collectively demonstrated the interaction between Pik_1_-H4 and Pik_2_-H4 and OsRRM2 in both yeast and in vitro assays.

Furthermore, we examined the expression profile of *OsRRM2* following challenge with the rice blast fungus in the susceptible cultivar Lijiangheituanxingu (LTH) and corresponding near-isogenic lines (NILs) carrying *Pik-H4* (referred to as *Pik-H4* NILs). In the absence of *Pik-H4*, *OsRRM2* was observed according to a fluctuating expression pattern and was not upregulated upon challenge with rice blast until 48 hpi (Appendix A). Notably, the relative expression levels of *OsRRM2* were significantly elevated at 12 h post-inoculation (hpi) in the *Pik-H4* NILs compared to LTH (Appendix A), while similar expression trends were observed at the rest of the time points, indicating the involvement of *OsRRM2* in *Pik-H4* NIL-mediated resistance.

### 2.2. OsRRM2 Promotes Pik_1_-H4 and Pik_2_-H4 Relocalization to the Vesicles and Chloroplasts

An analysis conducted using the UniProt database predicted the subcellular localization of OsRRM2 in the chloroplasts. To validate this prediction, we conducted subcellular localization assays in rice leaf sheath protoplast cells by fusing green fluorescent protein (GFP) with OsRRM2 for visualization. Our results indeed confirmed the chloroplast localization of OsRRM2-GFP, with the majority of the protoplasts (22 out of 30) exhibiting signals consistent with chloroplast localization. However, 8 out of 30 protoplasts displayed vesicular behavior (Appendix A), indicating the translocation of OsRRM2 between the vesicles and chloroplasts.

Since Pik_1_-H4 and Pik_2_-H4 (synonymous with Pikh-1 and Pikh-2, respectively) have been shown to localize in both the nucleus and cytoplasm [30] (Appendix A), clarifying the precise subcellular localization of the interactions between OsRRM2 and Pik_1_-H4 and OsRRM2 and Pik_2_-H4 was essential. To achieve this, we conducted biomolecular fluorescence complementation (BiFC) assays with the c-terminal of yellow fluorescent protein (YFPc)-OsRRM2, the n-terminal of YFP (YFPn)-Pik_1_-H4, and YFPn-Pik_2_-H4, along with the endosomal vesicle marker ARA6–red fluorescent protein (RFP) treated with wortmannin [31]. YFPc-OsRRM2 and YFPn-Pik_1_-H4 were observed within and around the vesicles in the protoplasts without (Figure 2a(top),b) and with (Figure 2a(bottom),c) chloroplasts, with weaker YFP signals also detected in the chloroplasts. A similar pattern was observed with YFPc-OsRRM2 and YFPn-Pik_2_-H4 (Figure 2d–f). Subsequent validation of the subcellular colocalization was performed using OsRRM2–cyan fluorescent protein (CFP) and Pik_1_-H4-GFP or Pik_2_-H4-GFP, which confirmed the observations made in the BiFC assays (Appendix A).

Given that *Pik* gene pairs are known to exhibit homo and hetero interactions [30,32], elucidating the interaction network involving OsRRM2, Pik_1_-H4, and Pik_2_-H4 was imperative. Therefore, we conducted fluorescence resonance energy transfer (FRET) assays by co-expressing OsRRM2-CFP, YFPn-Pik_1_-H4, and YFPn-Pik_2_-H4 in the protoplasts. Notably, the excitation of YFP fluorescence by OsRRM2-CFP occurred within chloroplast-like organelles (Appendix A), indicating the formation of an interaction complex involving OsRRM2, Pik_1_-H4, and Pik_2_-H4.

In summary, our findings suggested that OsRRM2 facilitated the translocation of Pik_1_-H4 and Pik_2_-H4 from the cytoplasm/nucleus to the endosomal vesicles and protoplasts by directly interacting with Pik_1_-H4 and Pik_2_-H4.

### 2.3. OsRRM2 Has a Minor Influence on Pik-H4-Related Resistance

To investigate the influence of *OsRRM2* on *Pik-H4*-mediated resistance, we generated *Pik-H4* NILs overexpressing *OsRRM2* or with *OsRRM2* knocked out (designated as OE-*OsRRM2*-GFP/*Pik-H4* NILs and ko-*OsRRM2*/*Pik-H4* NILs, hereafter referred to as OE-*OsRRM2* and ko-*OsRRM2*, respectively) (Appendix A). Transgenic lines exhibiting high or low relative expression levels of *OsRRM2* were subsequently selected for further analysis (Appendix A). In addition, we failed to generate transgenic LTH plants because of the multitudinous microorganisms that the LTH seeds carried.

Rice blast punch inoculation assays were conducted using 6- to 8-week-old plants (Figure 3a–c). Intriguingly, the plants with low *OsRRM2* expression displayed significantly enlarged lesion lengths compared to the *Pik-H4* NILs (*p* = 0.0017), albeit slightly lower than those observed in the OE-*OsRRM2* plants (Figure 3a,b, *p* = 0.0309). However, *OsRRM2* did not exhibit any discernible effect on rice blast resistance in terms of the spore ratio of the lesion areas (Figure 3c).

To elucidate the mechanism underlying *OsRRM2*’s impact on rice blast resistance, we analyzed the fold change in the expression levels of disease resistance genes and reactive oxygen species (ROS)-related genes (*p* < 0.05, |log_2_(Foldchange)| ≥ 2). The transcript levels of the disease resistance genes *PR-1a*, *PR-1b*, and *PBZ1* were found to be reduced more than twofold in the ko-*OsRRM2* plants (Figure 3d), whereas no significant differences were observed in the expression of the ROS-related genes (Figure 3e). The transcriptome analysis yielded results similar to those for the relative gene expression levels (Appendix A). These findings contributed partially to explaining the slight attenuation observed upon depleting *OsRRM2* in *Pik-H4*-mediated resistance.

### 2.4. OsRRM2 Mediates Pik-H4 Resistance via Chloroplast Immunity Pathways

The concept of immunity in the chloroplasts has gained attention in terms of plant disease resistance [33,34,35]. Given the localization of OsRRM2 in the chloroplasts, we hypothesized its involvement in chloroplast immunity. To test this hypothesis, we investigated the subcellular localization of OsRRM2-GFP in rice leaf sheaths challenged with rice blast fungus. However, the subcellular transport of OsRRM2 did not exhibit specific alterations (Appendix A). Consequently, we redirected our focus towards chloroplast-immunity-related pathways and genes through transcriptome analysis [*p* < 0.05, |log_2_(Foldchange)| ≥ 2]. We focused on the chloroplastic fatty synthesis pathways that participate in the biosynthesis of the resistance-related hormones jasmonic acid (JA) and salicylic acid (SA). As shown in Figure 4, *EG1* (*p* = 0.1142, log_2_Foldchange = 6.4865) and *ICS1* (*p* = 0.0038, log_2_Foldchange = 0.7787) were upregulated in the OE-*OsRRM2* lines, while more genes were downregulated in the ko-*OsRRM2* lines. This observation might elucidate the mild decline in resistance observed in the ko-*OsRRM2* plants in the punch inoculation assays.

Among the upregulated differentially expressed genes (DEGs), *LHCB5* [36] and *OsAPX8* [37], known to participate in chloroplast immunity by upregulating ROS, exhibited a slight increase compared to in the *Pik-H4* NILs with |log_2_(Foldchange)| ≤ 1 (Appendix A). Although the expression levels of *LHCB5* and *OsAPX8* were significantly higher in the ko-*OsRRM2* lines (Appendix A), the numerical differences in their expression were minor. This finding suggested that *OsRRM2* primarily influenced chloroplast immunity via chloroplastic fatty acid pathways rather than ROS pathways, consistent with the results in Figure 3e.

In summary, the changes in the expression levels of *OsRRM2* altered the chloroplast immunity and plant–pathogen interaction pathways in the *Pik-H4* NILs, indicating the participation of *OsRRM2* in *Pik-H4* resistance.

### 2.5. OsRRM2 Promotes RNA Editing of NdhB

Since OsRRM2 was not localized in the nucleus, we conducted a phylogenetic analysis of rice RRM family proteins to explore its putative role. Surprisingly, the results revealed that OsRRM2 did not cluster with any other reported rice RRM protein, even displaying considerable genetic distance (Appendix A). Consequently, we shifted our focus to investigating the putative RNA-binding ability of OsRRM2 to elucidate its mechanism in *Pik-H4*-mediated rice blast resistance. We performed systematic evolution of ligands by exponential enrichment (SELEX) to identify the RNA-binding sites for OsRRM2. After five rounds of SELEX, three sequences were significantly enriched (Figure 5a), and a sequence logo representing the OsRRM2-binding sites was generated (Figure 5b) following alignment.

Subsequently, we investigated the single-nucleotide polymorphisms (SNPs), alternative splicing (AS), and fold changes in the chloroplast-located genes (*p* < 0.05) using *OsRRM2* transcriptome data against the identified binding site 5′-WSTTMATCARG-3′. Remarkably, only one gene, *ndhB*, exhibited a C-to-T editing event at locus 8430 with a frequency of 0.3127 in the chloroplast genome (exclusive of the nucleus or mitochondrial genome) of the OE-*OsRRM2* lines. *ndhB* encodes a subunit of the NADH dehydrogenase complex, located on the plastid membrane, and plays a crucial role in NADH oxidation [38]. Our findings suggested that *OsRRM2* promoted RNA editing of chloroplastic *ndhB*, potentially exerting a minor effect on *Pik-H4*-mediated rice blast resistance through oxidation in the chloroplasts.

## 3. Discussion

As a major protein family in ETI, NLRs play a pivotal role in defending plants against pathogens. NLRs orchestrate transcriptional reprogramming by interacting with TFs, underscoring their significance in regulating disease-related genes [39]. It is noteworthy that cytoplasmic and nuclear NLRs function as sensors for pathogens and contribute to the regulation of disease-related genes. Additionally, certain NLRs, particularly those of the TNLs, have been shown to function as Ca^2+^ channels or NAD enzymes within a complex termed the resistosome, further illustrating the multifaceted roles of NLRs in ETI [8,9,40,41].

NLRs execute ETI by interacting with various proteins, among which the CC domain of CNLs is extensively involved in disease signal transduction and sensing processes [13,28,29]. For instance, the CNL-type rice blast resistance gene pair *Pik-H4* confers broad-spectrum [24] resistance to rice blast and mediates this resistance through its interaction with the TF OsBIHD1 [28]. In the context of this study, we identified an RRM protein, OsRRM2, which interacted with both Pik_1_-H4 and Pik_2_-H4, thereby facilitating their localization to the chloroplasts. Our findings suggested that OsRRM2 was intricately involved in the *Pik-H4*-related chloroplast immunity pathways.

Serving as key Avr sensors, the majority of NLRs are typically situated in the cytoplasm and nucleus, where they detect Avr proteins and transduce signals for disease resistance [13,30,42,43]. Notably, two CNL proteins have been documented to localize within the vesicles, suggesting a broader diversity in their subcellular localization and functions. In rice, the CNL protein Pib forms a complex with the SH3 protein SH3P2 within the vesicles, preventing an HR from being triggered by SH3P2 in the absence of AVRPib. However, in the presence of AVRPib, weakening of the interaction between SH3P2 and Pib results in the dissociation of Pib from the complex and the formation of dimers, which serve as functional units mediating the HR response [14]. Similarly, in maize, the CNL protein Rp1-D21 can induce the HR phenotype even under non-pathogen-invasive conditions, exhibiting localization in both the nucleus and cytoplasm. Interaction with ZmVPS23L, an ESCRT-related protein primarily localized within the vesicles, alters the subcellular localization of Rp1-D21, translocating it from the nucleoplasm to the vesicles [15].

In the case of the *Pik-h* gene pair, which is reported to localize in both the nucleus and cytoplasm, consistent with the corresponding AvrPikh [30], our findings revealed interactions between OsRRM2 and both Pik_1_-H4 and Pik_2_-H4 within the endosomal vesicles in cells without chloroplasts (Figure 2), indicating the OsRRM2-mediated transportation of Pik_1_-H4 and Pik_2_-H4. These interactions were found in vesicles and protoplasts in normal protoplast cells with weaker signals. These interactions suggest its potential role in facilitating the transport of Pik_1_-H4 and Pik_2_-H4 to the chloroplasts via vesicular transport mechanisms. Our results highlighted the involvement of the CNL pair *Pik-H4* in the chloroplasts. Additionally, transcriptome analysis revealed the enrichment of pathways such as the cellular skeleton (Appendix A), indicating that NLRs might exert diverse effects on the immune response by facilitating vesicular transport between various organelles.

ETI can indeed exert an influence on chloroplast immunity, as evidenced by studies demonstrating the involvement of chloroplasts in the immune response mediated by NLRs. In potatoes, for instance, the CNL protein Rpi-vnt1.1 triggers an immune response upon encountering AVRvnt1, secreted by *Phytophthora infestans*, and this response is regulated by light. AVRvnt1 can interact with the nuclear-encoded chloroplast protein GLYK, resulting in the activation of Rpi-vnt1.1 [16]. Notably, when infected by *Phytophthora infestans* in the dark, GLYK produces shorter transcripts lacking domains that interact with AVRvnt1, and both splice variants of GLYK are regulated by light. Similarly, in *Arabidopsis*, two protein kinases, MPK3/MPK6, induced by *Pseudomonas syringae*, are rapidly activated in response to AvrRpt2. This activation leads to the inhibition of photosynthesis, resulting in ROS accumulation. MPK3/MPK6 are essential to the immune responses mediated by the CNL protein RPS2 and the TNL protein RPS4 [44].

Moreover, a study has demonstrated that the tobacco CNL protein N’ interacts with the chloroplast-localized vesicle formation protein THF1 in the cytoplasm. During the immune response, N’ mediates the degradation of the THF1 protein, weakening the hypersensitive response [17]. Our findings, showing the relocation of Pik_1_-H4 and Pik_2_-H4 in the chloroplasts facilitated by OsRRM2 (Figure 2), further support the notion that these proteins participate in chloroplast immunity. In the OE-*OsRRM2* plants, pathways related to chloroplast immunity [45], such as fatty acid biosynthesis, were proportionately upregulated (Figure 4). However, the ROS-related chloroplast immunity genes *LHCB5* [36] and *OsAPX8* [37] did not exhibit differential expression in the OE-*OsRRM2* and ko-*OsRRM2* lines (Figure 4). These results suggest that *Pik-H4*, in collaboration with *OsRRM2*, re-programmed the chloroplastic fatty acid metabolism to enhance disease resistance. Nevertheless, the specific mechanisms underlying the functioning of the CNL gene pair *Pik-H4* in the chloroplasts during infection remain to be elucidated.

As a chloroplast-localized RNA-binding protein, OsRRM2 likely participates in various chloroplast functions, including splicing, RNA editing, and potentially other regulatory mechanisms. Notably, the classic RRM protein PIBP1 accumulates in the nuclear location of the CNL PigmR through direct interaction. Intriguingly, PIBP1 acts as a TF, binding directly to the promoters of disease-resistance genes and enhancing transcription [13]. Initially, our hypothesis posited similarities between the mechanisms of *Pik-H4* and *OsRRM2* and those of *Pigm* and *PIBP1*. However, analysis of the subcellular localization of OsRRM2 revealed the absence of nucleus-targeted TF capabilities (Appendix A). Phylogenetic analysis further demonstrated a significant genetic distance between *OsRRM2* and other reported members of the rice RRM family genes (Appendix A).

Efforts were undertaken to investigate disease resistance and photosynthetic phenotypes, revealing either minor differences (Figure 3a) or comparable performance (Appendix A) upon the overexpression or depletion of *OsRRM2*. Our findings suggested the modest impact of *OsRRM2* on *Pik-H4* resistance, indicating redundancy between *OsRRM2* and its orthologous genes, as delineated in the phylogenetic tree. These observations underscore the polymorphism inherent in NLR mechanisms.

The SELEX experiments identified potential binding sequences for OsRRM2 (Figure 5a), while RNA-seq analysis revealed OsRRM2’s facilitation of RNA editing in the NADH dehydrogenase complex member *ndhB*. This complex played a crucial role in generating oxidized NAD^+^ through redox reactions. Consequently, it was inferred that OsRRM2, through modulation of *ndhB*, influenced chloroplast-mediated ROS production, thus potentially executing diverse functions within the chloroplasts.

Our findings unveiled a potential chloroplast immune pathway wherein OsRRM2 co-localized with Pik_1_-H4 and Pik_2_-H4 in the chloroplasts, offering novel insights into the co-functionality of NLRs and RRMs in chloroplast immunity. Furthermore, owing to its RNA-binding capability, *OsRRM2* might play a role in mediating chloroplastic pathways related to fatty acids, although we have yet to elucidate its specific involvement in *Pik-H4*-mediated resistance. Given the involvement of *OsRRM2* and *PIBP1* in NLR-mediated resistance, there exists a promising avenue for the rational design of RRM proteins aimed at enhancing plant disease resistance.

## 4. Materials and Methods

### 4.1. Strains and Plasmid Construction

*E. coli* DH5α was utilized for the plasmid construction, whereas *E. coli* BL21 was employed for the protein expression. The Y2HGold yeast strain from the TAKARA Matchmaker Gold Yeast Two-Hybrid System was selected for the Y2H assays. Additionally, a *Magnaporthe oryzae* strain, GDYJ7 [28], harboring AvrPik-h, was cultured and utilized for inoculation purposes.

All the plasmids utilized in this study were constructed using the homologous recombination method using theClonExpress II One-Step Cloning Kit (Vazyme Biotech Co., Ltd., Nanjing, China). For instance, the Y2H plasmid pGADT7-OsRRM2 was generated by recombining the OsRRM2 fragment flanked by recombination arms with pGADT7 and linearized pGADT7 using the appropriate restriction endonuclease. Subsequently, the recombination product was transformed into *E. coli* DH5α using the heat shock method. Following colony PCR, next-generation sequencing, and alignment, the appropriate strain was stocked, and the plasmid was extracted for subsequent experiments. All the primers utilized in this study are listed in Appendix A.

### 4.2. Plant Materials and Growth Conditions

For protoplast isolation and transformation, Nipponbare (Nip) seeds were subjected to sterilization with 75% ethanol and 15% NaClO and then grown on an MS medium. The rice-blast-susceptible rice line LTH served as the negative control. The rice-blast-resistant line *Pik-H4* NILs/LTH (*Pik-H4* NILs), carrying *Pik-H4* into LTH, was previously constructed. Overexpression of *OsRRM2* (OE-*OsRRM2*) was achieved by transforming pOx-P_Ubi_-*OsRRM2*-*GFP* into the *Pik-H4* NILs. The depletion of *OsRRM2* (ko-*OsRRM2*) was brought about by targeting the initiation codon of *OsRRM2* using the CRISPR/Cas9 system in the *Pik-H4* NILs. The transgenic plants were PCR-tested, and F_2_-positive plants were selected for generation and phenotype analysis. The experimental plants were cultivated in a field designated for transgenic plants. The inoculation assays were conducted in a growth chamber with a temperature cycle of 30 °C/light and 26 °C/dark.

### 4.3. Yeast Two-Hybrid and GST-Pulldown Assays

The Y2H assays were conducted in accordance with the manufacturer’s instructions. Briefly, the protein-coding sequences targeted for interaction were fused with the GAL4 activation domain (AD) or binding domain (BD) and cloned into the Y2H plasmids pGADT7 and pGBKT7, respectively (e.g., pGADT7-*OsRRM2* and pGBKT7-*Pik_1_-H4*). These Y2H plasmids were co-transformed into the Y2HGold strain and selected on SD/-Leu/-Trp (double-dropout, DDO) medium. Confirmation of the interaction was performed on an SD/-Leu/-Trp/-His/Ura (Quadra dropout, QDO) medium.

The in vivo pulldown assays were conducted overnight using GST-tagged and His-tagged proteins extracted from *E. coli* BL21, induced with isopropyl-beta-D-thiogalactopyranoside (IPTG). Approximately 30 μg of GST-OsRRM2, along with Pik_1_-H4-CC-His or Pik_2_-H4-CC-His (~10 μg), was incubated with α-GST beads (GST-tagged protein purification bead kit, Beyotime Biotech. Inc., Shanghai, China) at 4 °C overnight with gentle agitation. Following washing the beads with elution buffer, the precipitates were boiled in protein-loading buffer for 5 min and analyzed using standard SDS-PAGE and immunoblotting techniques.

### 4.4. Subcellular Localization and Bimolecular Fluorescence in the Protoplasts

Protoplasts were isolated from the rice leaf sheaths as previously described [46]. Chloroplast localization was achieved using green leaf sheaths. The protocol was modified by incubating the resuspended cells in MMG solution on ice for 30 min and washing the PEG-treated cells twice with W5 solution. Up to 20 ng of plasmid DNA was added to the protoplasts for the subcellular localization studies. For the co-localization and BiFC experiments, approximately 10 ng of each plasmid was mixed with the cells. After 12–16 h, the protoplasts were imaged using a confocal microscope (LSM 750, Carl Zeiss, Oberkochen, Germany), and the images were processed using ZEN 3.6 (Carl Zeiss, Oberkochen, Germany). Wortmannin at a concentration of 33 μm was added to the protoplasts for visualization of the vesicle marker ARA6, as previously reported [31]. The excitation/emission wavelengths used were 488 nm/490–560 nm for GFP, 516 nm/529–550 nm for YFP, 543 nm/580–660 nm for RFP, and 640 nm/670 nm for chloroplast autofluorescence.

### 4.5. Rice Blast Inoculation

Spray inoculation was conducted as previously described [47], with modifications to assess the gene expression levels. The spores were collected using 0.02% Tween 20, and the spore concentration was adjusted to 5 × 10^5^ spores/mL. The spore suspension was evenly sprayed onto the surface of 2-week-old rice seedling leaves until visible droplets formed using a spray gun. Samples were collected at 0, 6, 12, 24, 36, 48, 60, and 72 hpi after 12 h of incubation in darkness for quantitative real-time PCR (qRT-PCR). For the phenotypic assessment, leaves of 6- to 8-week-old seedlings were punch-inoculated, as described in [48]. The leaf sheath inoculation was performed as previously reported, with a spore concentration of 5 × 10^5^ spores/mL. Disease symptoms were observed at 7 days post-inoculation (dpi), and the lesion length was determined using the ImageJ software (https://imagej.net/ij/, accessed on 11 May 2024). The sporulation rate on the lesions was calculated following established protocols [48].

### 4.6. RNA Extraction, Reverse Transcription PCR, and qRT-PCR

Total RNA was extracted from the inoculated leaves using TRIzol reagent (Thermo Fisher Scientific Inc., Walthm, MA, USA) according to the manufacturer’s instructions. The RNA samples were reverse-transcribed into cDNA using Vazyme HiScript III All-in-one RT SuperMix Perfect for qPCR. The qRT-PCR reaction systems were prepared following the manufacturer’s instructions (AceQ Universal SYBR qPCR Master Mix, Vazyme Biotech Co., Ltd., Nanjing, China) using a Real-Time PCR System (StepOnePlus™, Applied Biosystems, Thermo Fisher Scientific Inc., Walthm, USA). The ΔΔCt method was employed to evaluate the gene expression levels, with rice *β-actin* (*LOC_Os03g50885*) serving as the control gene. Three biological replicates were performed for the qRT-PCR analysis.

### 4.7. Phylogenetic Analysis

To investigate the rice RRM gene family, the RRM domain Hidden Markov Model (HMM) was obtained from Pfam and utilized to screen the RRM genes in the rice genome (threshold: 10^−60^) to establish a rice-specific HMM. Subsequently, this new HMM was employed to search the rice genome (threshold: 10^−4^). The identified RRM protein sequences were aligned using ClustalW (https://www.genome.jp/tools-bin/clustalw, accessed on 11 May 2024), and a phylogenetic tree was constructed using the Maximum Likelihood method with the default settings via MEGA11.

### 4.8. Transcriptome Analysis

To elucidate the putative transcriptional function of *OsRRM2*, transcriptome profiling was conducted through RNA-seq. Shoots of the 3- to 4-week-old *Pik-H4* NILs, OE-*OsRRM2*, and ko-*OsRRM2* were sampled. Total RNA extraction was carried out using TRIzol reagent (Invitrogen), and mRNA enrichment was achieved using Oligo (dT) beads. Subsequently, the enriched mRNA was fragmented into short fragments using a fragmentation buffer and reverse-transcribed into cDNA using the NEBNext Ultra RNA Library Prep Kit for Illumina (New England Biolabs Inc., Ipswich, MA, USA). The resulting double-stranded cDNA fragments underwent end repair, A-base addition, and ligation to Illumina sequencing adapters. The ligation reactions were purified using AM-Pure XP Beads (1.0×, followed by PCR amplification. The resulting cDNA library was sequenced using Illumina NovaSeq 6000 from Gene Denovo Biotechnology Co. (Guangzhou, China). DEGs were identified using a significance threshold of *p* < 0.01 and |log_2_Foldchange| ≥ 2. Correlation analysis was conducted using R software (https://www.r-project.org/, accessed on 11 May 2024) to assess the reliability and operational stability of the experimental results. Calculation of the correlation coefficient between two replicates provided an evaluation of the repeatability between samples.

### 4.9. Systematic Evolution of Ligands by Exponential Enrichment

SELEX was conducted with modifications to identify the potential RNA-binding sites of OsRRM2. Initially, SELEX-Oligo sequences containing 40 random nucleotides were designed, flanked with homologous arms of pUC19, and incorporated with the *Hin*dIII and *Bam*HI restriction enzyme sites. Random sequences were PCR-amplified using the SELEX-F (P_T7_-TSS_T7_-homologous arm-*Hin*dIII) and SELEX-R (*Bam*HI-homologous arm) primers, followed by gel extraction. The purified PCR products were transcribed in vitro using the T7 High Yield RNA Transcription Kit (Vazyme Biotech Co., Ltd., Nanjing, China), and the transcribed products were purified using phenol/chloroform extraction. The RNA quality was assessed using gel electrophoresis and UV spectrophotometry.

Next, the RNA, His-tagged OsRRM2 protein, and anti-His beads were incubated overnight at 4 °C, and the RNA–protein complexes were purified according to the manufacturer’s instructions. The purified RNA was reverse-transcribed, and the reverse transcription products were amplified using the SELEX-F and SELEX-R primers. These amplification products served as the template for the subsequent round of SELEX, which was repeated for five rounds of selection. Finally, the amplified products were recombined into pUC19 plasmids using homologous recombination. Positive colonies were selected via blue/white screening, followed by colony PCR and Sanger sequencing for validation.

## 5. Conclusions

In this study, the RRM protein OsRRM2, located in the vesicles and chloroplasts, was found to interact with the NLR pair Pik_1_-H4/Pik_2_-H4 and mediated the translocation of the NLR pair to the vesicles and chloroplasts. Knocking out *OsRRM2* in the *Pik-H4* NILs slightly impaired the rice blast resistance. The pathogenesis-related gene *PR10*, a chloroplastic fatty acid synthesis pathway gene that contributes to JA and SA biosynthesis, was upregulated when *OsRRM2* was overexpressed. A putative OsRRM2-binding motif and an RNA (*ndhb*) were screened through SELEX and RNA-seq analysis. Altogether, the results suggest *OsRRM2* is involved in *Pik-H4* mediated resistance through direct interaction with the pair and translocates the NLR pair to the chloroplasts. Furthermore, *OsRRM2* promotes chloroplastic immune pathways.

## Figures and Tables

**Figure 1 ijms-25-05557-f001:**
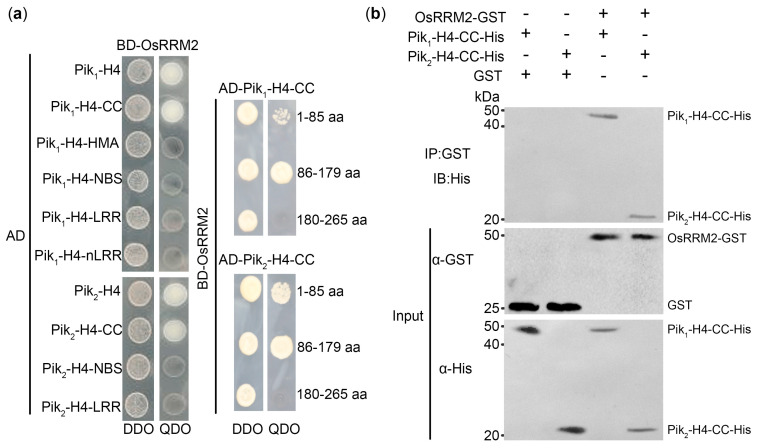
OsRRM2 interacts with Pik_1_-H4 or Pik_2_-H4. (**a**) Split Y2H interaction of OsRRM2 with Pik_1_-H4 or Pik_2_-H4; (**b**) GST pulldown assays of OsRRM2-GST with Pik_1_-H4-CC-His or Pik_2_-H4-CC-His. Activation domain (AD); binding domain (BD); coiled-coil (CC); heavy-metal-associated (HMA); nucleotide-binding site (NBS); leucine-rich repeat (LRR); SD/-Leu/-Trp (double-dropout, DDO) medium; SD/-Leu/-Trp/-His/Ura (Quadra dropout, QDO); medium glutathione S-transferase (GST).

**Figure 2 ijms-25-05557-f002:**
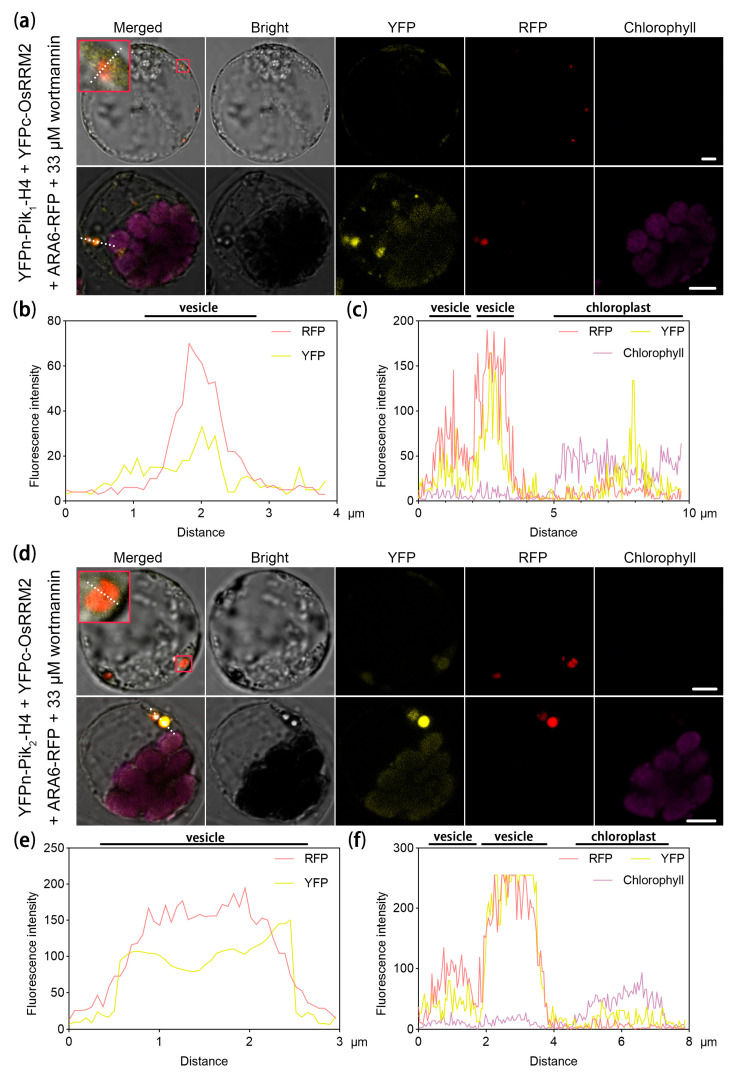
OsRRM2 interacts with Pik_1_-H4 or Pik_2_-H4 in vesicles and chloroplasts. (**a**,**d**) Interaction of YFPn-Pik_1_-H4 [YFPn-Pik_2_-H4 in (**d**)] and YFPc-OsRRM2 in protoplasts with/without chloroplasts. The red boxes show a partially enlarged image of the co-location of YFP and RFP signals. (**b**,**c**,**e**,**f**) The fluorescence distribution indicating the co-location events in (**a**) or (**d**); the fluorescence intensity was measured along the white dot lines in (**a**,**d**). The BiFC assays were performed in albino seedlings without matured chloroplasts [top of (**a**,**d**)] and protoplasts with chloroplasts [bottom of (**a**,**d**)]; ARA6-RFP with wortmannin was used as an endosomal vesicle marker. YFPc: c-terminal of yellow fluorescent protein. YFPn: n-terminal of YFP. RFP: red fluorescent protein. The ARA6-RFP was co-expressed with wortmannin to indicate the vesicles. Scale bars = 5 μm.

**Figure 3 ijms-25-05557-f003:**
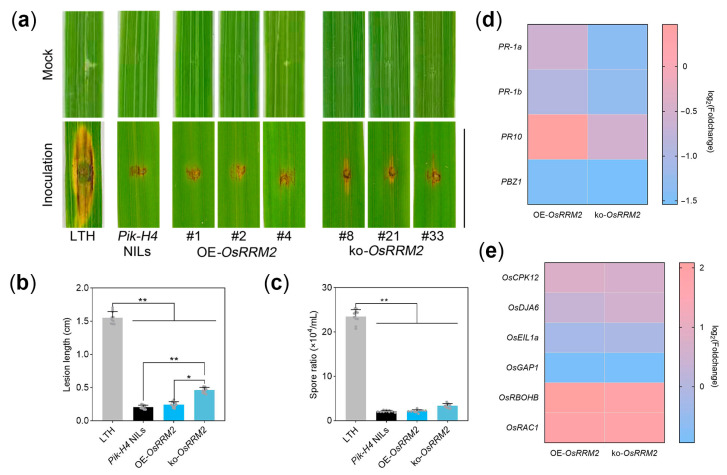
Rice blast resistance analysis of OsRRM2. (**a**) Punch inoculation assays for phenotypes of LTH, *Pik-H*4 NILs, OE-*OsRRM2*, and ko-*OsRRM2*. Scale bar = 1 cm. (**b**,**c**) The lesion lengths and spore ratios of lesion areas in (**a**). Three lines of overexpression or knockout of *OsRRM2* were measured. The values are shown as means ± SD (n = 10). Asterisks indicate a significant difference calculated using Student’s *t*-test (* *p* < 0.05, ** *p* < 0.01). (**d**,**e**) The fold change in resistance-related genes and ROS-related genes of OE-*OsRRM2* and ko-*OsRRM2*.

**Figure 4 ijms-25-05557-f004:**
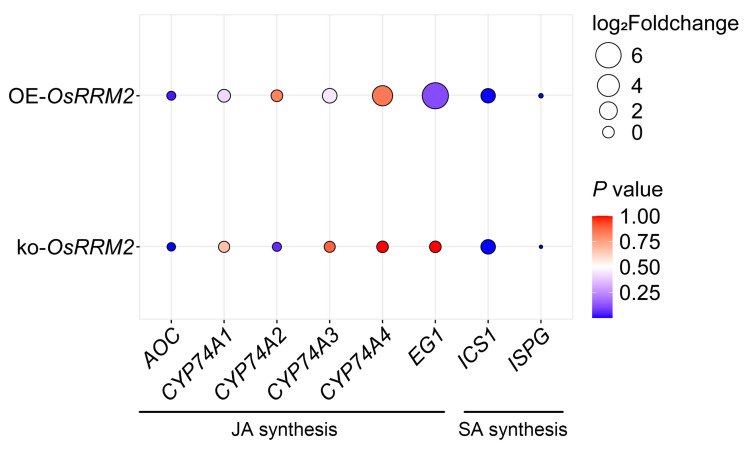
The fold change in chloroplastic fatty synthesis pathway genes related to jasmonic acid (JA) and salicylic acid (SA) in *OsRRM2* plants.

**Figure 5 ijms-25-05557-f005:**
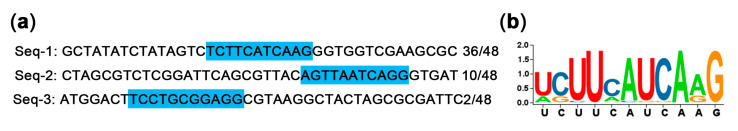
RNA-binding site analysis of OsRRM2. (**a**) Three enriched sequences from the SELEX assay. The binding sites are shaded in blue. The numbers represent the number of sequences out of the total positive strains from the fifth enrichment; (**b**) the sequence logo of the OsRRM2-binding site.

## Data Availability

The raw RNA-seq data have been deposited into the Genome Sequence Archive database (accession number CRA015700) of the China National Center for Bioinformation.

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
