# Peer review of "Unveiling the Role of RNA Recognition Motif Proteins in Orchestrating Nucleotide-Binding Site and Leucine-Rich Repeat Protein Gene Pairs and Chloroplast Immunity Pathways: Insights into Plant Defense Mechanisms"

_ijms, 2024, doi:10.3390/ijms25105557_

Round 1

Reviewer 1 Report

Comments and Suggestions for Authors

Major comments:

The manuscript by Gu et al. identified the RNA recognition motif protein OsRRM2 as a key enhancer of rice blast resistance in plants. The authors claimed that OsRRM2 interacts with the NLR gene pair Pik-H4 within chloroplasts, boosting resistance-related genes and promoting RNA editing of the chloroplastic gene ndhB. These findings revealed OsRRM2's role in linking plant immunity and chloroplast defense, deepening our understanding of plant immune mechanisms. Overall, the manuscript describes some important findings that are of interest to a broad audience.

  1. How does OsRRM2 promote RNA editing in the chloroplastic gene ndhB? Is there a direct interaction with the RNA or does it involve other mediators?
  2. Have similar RRM proteins been identified in other plant species or in relation to other diseases? How does OsRRM2 compare to these in terms of structure and function? 
  1. Could the authors elaborate on the pathways related to chloroplast immunity that are enhanced by OsRRM2? How do these pathways specifically contribute to resistance against rice blast disease? 
  1. Do the authors have data showing the differences in OsRRM2 protein levels between OE-OsRRM2 (overexpression) plants and ko-OsRRM2 (knockout) plants?

Minor comments: 

  1. The reference format does not align with the guidelines provided by the journal of interest. 
  2. I recommend moving Supplementary Figures 1-7 to the main text.
  3. Line 465, GSA stands for Genome Sequence Archive?
  4. Line 466, CRA015700) from China National Center for Bioinformation is so far not accessible.
Comments on the Quality of English Language

please see above. 

Author Response

1. Summary

Comments 1: How does OsRRM2 promote RNA editing in the chloroplastic gene ndhb? Is there a direct interaction with the RNA or does it involve other mediators?

Response 1: Overexpressing OsRRM2 promotes the RNA editing of chloroplastic ndhb by C to T changes. Our result was analyzed through RNA-seq and was not confirmed by the RNA-interaction experiment.

Comments 2: Have similar RRM proteins been identified in other plant species or in relation to other diseases? How does OsRRM2 compare to these in terms of structure and function?

Response 2: Some reports have found that RRM proteins participate in immunity. For example, AtGRP7 is ADP-ribosylated by Pseudomonas syringae pv. tomato DC3000 effector HopU1 within its RRM domain (doi:10.1038/nature05737). GRP7 is located in cytoplasm, nucleus and extracellular vesicles (doi:10.1093/plcell/koac043) and involves in alternative splicing of circadian genes (doi: 10.1111/j.1365-313X.2007.03302.x). AtCPR5, a negative immune regulator of Pseudomonas syringae pv. maculicola ES4326, is a member of an alternative splicing complex and is located in nuclear speckles binding RNAs of CPR5-regulated alternatively spliced genes (doi:10.1093/plcell/koac037). GRP7 and CPR5 have one RRM domain and are involved in alternative splicing, while OsRRM2 has two. These reports are supplemented in lines 65-66. In our RNA-seq results, alternative splicing of nuclear genes (no chloroplastic genes) was observed. Given the chloroplastic location of OsRRM2, we did not study these nuclear alternative splicing events. In our previous study on OsRRM2, only one chloroplastic RNA edit event (ndhb) was found.

Comments 3: Could the authors elaborate on the pathways related to chloroplast immunity that are enhanced by OsRRM2? How do these pathways specifically contribute to resistance against rice blast disease?

Response 3: OsRRM2 enhanced lipid metabolism pathways, including fatty acid elongation, cutin, suberine and wax biosynthesis, fatty acid metabolism, glycerolipid metabolism, fatty acid degradation, and fatty acid biosynthesis. The salicylic acid precursor isochorismate, the stearic that participates in jasmonic acid synthesis, salicylic acid synthesis, digalactosyldiacylglycerol, which is required for pathogen-induced biosynthesis of nitric oxide, the nitric oxide triggers reactive oxygen species; these chemicals are derived from the chloroplast (doi: 10.1146/annurev-phyto-020620115813).

Comments 4: Do the authors have data showing the differences in OsRRM2 protein levels between OE-OsRRM2 (overexpression) plants and ko-OsRRM2 (knockout) plants?

Response 4: Although overexpression of OsRRM2 was tagged with GFP, the knockout of OsRRM2 was carried out by targeting genomic DNA with CRISPR/Cas9 system. Therefore it is not available to detect the ko-OsRRM2 proteins. We only investigated the OsRRM2 expression levels through qRT-PCR.

Comments 5: The reference format does not align with the guidelines provided by the journal of interest.

Response 5: So sorry for our carelessness. We have fixed the reference format in the revised manuscript.

Comments 6: I recommend moving Supplementary Figures 1-7 to the main text.

Response 6: Inserting 7 figures to the main text will make the layout complicated. Combining with other reviewers’ comments, we deleted two supplementary figures. Thus we did not insert more figures to main text.

Comments 7: Line 465, GSA stands for Genome Sequence Archive?

Response 7: Yes. The full name, Genome Sequence Archive, is used in revised manuscript (Lines 489-490) in case of misunderstanding.

Comments 8: Line 466, CRA015700) from China National Center for Bioinformation is so far not accessible.

Response 8: So sorry for our carelessness, it is now available.

Reviewer 2 Report

Comments and Suggestions for Authors

The paper is interesting as identifies a novel protein-protein interaction, albeit is not really clear whether is significative or defining any key interaction. The main problem is that there are two figues that are very weak and require major improvement:

Figure 2: Nothing can be seen in the chlorophyll pannel, so the localization is very confuse. Enhance the figure to convinve that the interaction happens where indicated in the text.Which vesicles do the ARA6 marker indicate? please mention it in the figure legend.

Figure 3: Authors only consider 1 overexpression ,ines, please, include at least three to check that the phenotype is coherent, and not due to an artifact of the gene insertion.

Minor points:

Figure 1: too narrow, and avoid the lettering on the pics. Enlarge the figure and put the labels on the side.

Lines 21-22: this sentence is redundant. Delete.

Line 80: spacing before results.

Line 177 and 178: Simplify the sentence. Too literary.

Line 181: "to our disappointment". This is a scientific paper, not a Shakespeare play. Avoid sentimentalism and subjective observations and keep attached to the results.    

Comments on the Quality of English Language

needs minor improvement

Author Response

1. Summary

Thank you very much for taking the time to review this manuscript. Please find the detailed responses below and the corresponding revisions highlighted changes in the re-submitted files. Considering all reviewers’ comments, great changes are made in Figure 2, Figure 4, and their corresponding statements in the main text. Abbreviations are added to each figure. A conclusion part is added to the main text. The reference format is also corrected.

Comments 1: Figure 2: Nothing can be seen in the chlorophyll pannel, so the localization is very confuse. Enhance the figure to convinve that the interaction happens where indicated in the text. Which vesicles do the ARA6 marker indicate? please mention it in the figure legend.

Response 1: Sorry for our obscure statement. Four BiFC assays were carried out in albinic seedlings; thus, the protoplasts lacked chloroplasts. The chloroplasts were marked in purple in the chloroplast-containing protoplasts. We stated this in the revised manuscript (Lines 136-138). The ARA6 marked vesicles are mentioned in line 140.

Comments 2: Figure 3: Authors only consider 1 overexpression ,ines, please, include at least three to check that the phenotype is coherent, and not due to an artifact of the gene insertion.

Response 2: Sorry for our ambiguous description. The phenotype analysis was measured from three different lines of overexpression or knockout of OsRRM2. The description is added in lines 172-173.

Comments 3: Figure 1: too narrow, and avoid the lettering on the pics. Enlarge the figure and put the labels on the side.

Response 3: Thank you for your suggestion. We have adjusted Figure 1.

Comments 4: Lines 21-22: this sentence is redundant. Delete.

Response 4: Sorry for our misstep. We have deleted the redundant sentence in lines 21-22.

Comments 5: Line 80: spacing before results.

Response 5: Sorry for our misstep. We have added the space in line 82.

Comments 6: Line 177 and 178: Simplify the sentence. Too literary.

Response 6: We simplified the sentence in lines 193-194.

Comments 7: Line 181: "to our disappointment". This is a scientific paper, not a Shakespeare play. Avoid sentimentalism and subjective observations and keep attached to the results.

Response 7: We have deleted the statement in line 196.

Reviewer 3 Report

Comments and Suggestions for Authors

In this study, authors aimed to unveil the role of RNA recognition motif proteins in orchestrating NLR gene pairs and chloroplast immunity pathways.

The study design is acceptable. Experimental design and analyses seems relevant.

The results of this study contain some valuable elements such as the authors i) identified the interaction of a rice RRM protein (referred as OsRRM2) with both Pik1-H4 and Pik2-H4, with observations of this interaction occurring within vesicles and chloroplasts; ii) demonstrated the upregulation of pathways related to chloroplast immunity in transgenic plants overexpressing OsRRM2; and iii) identified the binding sites of OsRRM2 and RNA editing events within chloroplast proteins.

The Discussion is acceptable. I suggest to prepare a separate Conclusion section, in which authors address the limitations of their study and briefly discuss future needs for this topic.

The number of references is acceptable. Some inconsistency, for example, journal names are sometimes abbreviated sometimes not.

Overall, the study contains valuable results that can be considered for possible publication after suitable revisions.

Other suggestions:

  • L8: very strange e-mail address: e.g. 99g999999ufw@gmail.com
  • L80: delete point before Results
  • L108: In the title give explanation for OsRRM2, Pik1-H4 and Pik2-H4, DDO, QDO GST … etc.
  • L142: Give explanation in the title for YFp, RFP and OsRRM2, Pik1-H4 and Pik2-H4.
  • L194: Give explanation in the title for OsRRM2, NIL, KEGG.
  • L230: Give explanation in the title for OsRRM2.
  • L331. Full name for E. coli. And ‘coli’ and not ‘Coli’.

Author Response

1. Summary

Thank you very much for taking the time to review this manuscript. Please find the detailed responses below and the corresponding revisions highlighted changes in the re-submitted files. Considering all reviewers’ comments, great changes are made in Figure 2, Figure 4, and their corresponding statements in the main text. Abbreviations are added to each figure. A conclusion part is added to the main text. The reference format is also corrected.

In this study, authors aimed to unveil the role of RNA recognition motif proteins in orchestrating NLR gene pairs and chloroplast immunity pathways.

The study design is acceptable. Experimental design and analyses seems relevant.

The results of this study contain some valuable elements such as the authors i) identified the interaction of a rice RRM protein (referred as OsRRM2) with both Pik1-H4 and Pik2-H4, with observations of this interaction occurring within vesicles and chloroplasts; ii) demonstrated the upregulation of pathways related to chloroplast immunity in transgenic plants overexpressing OsRRM2; and iii) identified the binding sites of OsRRM2 and RNA editing events within chloroplast proteins.

Overall, the study contains valuable results that can be considered for possible publication after suitable revisions.

Comments 1: The Discussion is acceptable. I suggest to prepare a separate Conclusion section, in which authors address the limitations of their study and briefly discuss future needs for this topic.

Response 1: Agree. The conclusion was added to the revised manuscript in line 461.

Comments 2: The number of references is acceptable. Some inconsistency, for example, journal names are sometimes abbreviated sometimes not.

Response 2: Agree. The References part was revised in the main text.

Comments 3: L8: very strange e-mail address: e.g. 99g999999ufw@gmail.com

Response 3: Agree. However, the ID has been used for more than 20 years and never changed.

Comments 4: L80: delete point before Results

Response 4: It is revised in the revised manuscript line 82.

Comments 5: L108: In the title give explanation for OsRRM2, Pik1-H4 and Pik2-H4, DDO, QDO GST … etc.

Response 5: The explanation of OsRRM2 is added in line 75, the same as below. GST was explained in line 92. The explanation of DDO and QDO are added in lines 116-118.

Comments 6: L142: Give explanation in the title for YFp, RFP and OsRRM2, Pik1-H4 and Pik2-H4.

Response 6: The explanation are added in lines 139-140.

Comments 7: L194: Give explanation in the title for OsRRM2, NIL, KEGG.

Response 7: The explanation of NILs is added in line 105. We removed the KEGG analysis from the main text.

Comments 8: L230: Give explanation in the title for OsRRM2.

Response 8: The explanation of OsRRM2 is added in line 75.

Comments 9: L331. Full name for E. coli. And ‘coli’ and not ‘Coli’.

Response 9: It is corrected in line 346.

Reviewer 4 Report

Comments and Suggestions for Authors

The authors in their manuscript entitled ‘Unveiling the Role of RNA Recognition Motif Proteins in Orchestrating NLR Gene Pairs and Chloroplast Immunity Pathways: Insights into Plant Defense Mechanisms” present the functional characterization of the OsRRM2 regarding its subcellular localization, interactions, and roles revealed through mutant analyses. My main comments are with regards the presentation of the results in figure and table formats and in certain cases with the significance of the results and what is claimed by the authors.

In specific:

1.        Please provide a full and very detailed description of all abbreviations in all figures. Each figure should be a standalone self-explanatory case. Otherwise, the reader must go back and forward to the text and materials and methods section to follow what is shown.

2.        In figure S1 the authors should comment on the expression observed in the Mock in LTH. Why is there a fluctuation of expression observed in the mock? Expression at 12 and 48 gets high up, falling at normal at 24 and 72 hours.

3.        In In figure S1 the expression pattern in inoculated NILs does not actually follow the comments in the manuscript text. The authors comment only on what is observed at 12 hours, however, not in the rest of what is shown in the figure S1. Please argument on the expression observed in both strains, for mock and inoculated samples, for all hours measured. Otherwise, there may be a selective conclusion deduced.

4.        Please explain for the first time in the manuscript text what NILs is (line 102).

5.        In figure 1a explain in the legend what is shown in the QDO interactions for HMA, NBS, LRR, etc. Are these weak interactions that are observed or plate background, for example? It would be for the reader ease to add a cross or dash in positive/negative interaction below or by the side of each circle.

6.        In figure 1b explain in detail what are the protein bands shown. It rather difficult to follow what each band represents. The guide map on the top with of the blots is of help but not much help.

7.        In figure S2a the authors claim that thre is also vesicle localization. However, this is not supported by a relative marker. Only by visual assumption (I suppose from the bright field) then put in circles in Merged format. Please justify your conclusion.

8.        On the other hand, in figure 2 the authors claim (lines 125-126) that there is chloroplast colocalization (weak signal), however this is not clearly shown. Chlorophyll images and FP signals do not seem to overlap except in the 3rd row in merged, however, this is still not clear. It seems that are vesicles. Please justify your conclusion

9.        I would suggest that figures 2 and S2 are presented in the main text, probably rearranged, since all this information is important. However, even in such a case the aforementioned justifications should be provided.

10.   The authors should comment on why OE and ko lines were not generated in a susceptible (normal?) background. It would be useful to know what the protein does or not in such a background irrespective Pik-H4. If they have also been generated, please provide the relevant results. If not, please comment on your decision apart of this stated in lines 146-148.

11.   Please comment on your statement in lines 169-173, in relation to figure 3d and figure S3e. The results do not seem to match regarding fold-changes between images and text description.

12.   It is difficult to follow the comparison stated in lines 184-190 as presented in a table and two supplementary figures S5 and S6. Please provide a concise table summarizing all data discussed. I apologize but I cannot make a conclusion to what you describe in the form the data are presented.

Author Response

1. Summary

Thank you very much for taking the time to review this manuscript. Please find the detailed responses below and the corresponding revisions highlighted changes in the re-submitted files. Considering all reviewers’ comments, great changes are made in Figure 2, Figure 4, and their corresponding statements in the main text. Abbreviations are added to each figure. A conclusion part is added to the main text. The reference format is also corrected.

Comments 1: Please provide a full and very detailed description of all abbreviations in all figures. Each figure should be a standalone self-explanatory case. Otherwise, the reader must go back and forward to the text and materials and methods section to follow what is shown.

Response 1: The explanations of abbreviations are added in lines 116-118, lines 139-140, lines 208-209.

Comments 2: In figure S1 the authors should comment on the expression observed in the Mock in LTH. Why is there a fluctuation of expression observed in the mock? Expression at 12 and 48 gets high up, falling at normal at 24 and 72 hours. In In figure S1 the expression pattern in inoculated NILs does not actually follow the comments in the manuscript text. The authors comment only on what is observed at 12 hours, however, not in the rest of what is shown in the figure S1. Please argument on the expression observed in both strains, for mock and inoculated samples, for all hours measured. Otherwise, there may be a selective conclusion deduced.

Response 2: A comment is added in lines 105-107 and line 110. Since the only difference between Pik-H4 NILs and LTH is the resistant Pik-H4 versus susceptible Pik-LTH in the Pik locus, we consider the expression fluctuation of OsRRM2 in mock assays innate. OsRRM2 was not up-regulated in 12 hpi due to the absence of Pik-H4. The up-regulation of OsRRM2 in 48 hpi against rice blast treatment in LTH or Pik-H4 NILs is also an innate immune expression pattern.

Comments 3: Please explain for the first time in the manuscript text what NILs is (line 102).

Response 3: The full name of NILs is explained in line 105.

Comments 4: In figure 1a explain in the legend what is shown in the QDO interactions for HMA, NBS, LRR, etc. Are these weak interactions that are observed or plate background, for example? It would be for the reader ease to add a cross or dash in positive/negative interaction below or by the side of each circle.

Response 4: The explanation of the abbreviations is added in line 88 and lines 116-118. The description of weak interaction in Y2H assays is supplemented in lines 99.

Comments 5: In figure 1b explain in detail what are the protein bands shown. It rather difficult to follow what each band represents. The guide map on the top with of the blots is of help but not much help.

Response 5: The remarks are added to figure 1b.

Comments 6: In figure S2a the authors claim that thre is also vesicle localization. However, this is not supported by a relative marker. Only by visual assumption (I suppose from the bright field) then put in circles in Merged format. Please justify your conclusion.

Response 6: Sorry for our ambiguous display. The vesicles are too small for observation. Partial enlarge images are added to figure S2a.

Comments 7: On the other hand, in figure 2 the authors claim (lines 125-126) that there is chloroplast colocalization (weak signal), however this is not clearly shown. Chlorophyll images and FP signals do not seem to overlap except in the 3rd row in merged, however, this is still not clear. It seems that are vesicles. Please justify your conclusion

Response 7: The YFP fluorescence was weaker in chloroplast. We displayed BiFC in protoplasts without chloroplast, thus those protoplasts are lack of chlorophyll signals. The description is supplemented in lines 136-138. We added fluorescence distribution curves to figure 2 for better co-location of markers and YFP signals.

Comments 8:  I would suggest that figures 2 and S2 are presented in the main text, probably rearranged, since all this information is important. However, even in such a case the aforementioned justifications should be provided.

Response 8: Agree. Our arrangement is based on: (1)For better display of vesicles (especially the small ones), we used larger figures; (2) Such many channels and figures are difficult to layout, thus we displayed the most important figures in the main text.

Comments 9: The authors should comment on why OE and ko lines were not generated in a susceptible (normal?) background. It would be useful to know what the protein does or not in such a background irrespective Pik-H4. If they have also been generated, please provide the relevant results. If not, please comment on your decision apart of this stated in lines 146-148.

Response 9: Agree. However, the susceptible LTH carries many native microorganisms and is difficult to sterilize. Indeed, we tried to generate LTH calluses and failed. We have also asked a biotech company to generate LTH transgenic plants. Since we chose OsRRM2 for research from a Y2H screening with Pik1-H4 and Pik2-H4, we did not spend time on LTH transgenic plants. A description is added in lines 165-166.

Comments 10: Please comment on your statement in lines 169-173, in relation to figure 3d and figure S3e. The results do not seem to match regarding fold-changes between images and text description.

Response 10: The expression trends matched between RNA-seq and qRT-PCR but with different fold-changes mainly because of the sampling tissues. Leaves were sampled for qRT-PCR and shoots for RNA-seq. In general, the expression trends are consistent.

Comments 11: It is difficult to follow the comparison stated in lines 184-190 as presented in a table and two supplementary figures S5 and S6. Please provide a concise table summarizing all data discussed. I apologize but I cannot make a conclusion to what you describe in the form the data are presented.

Response 11: Sorry for our confusing visualization. For better understanding, we substituted the table with Figure 4, which shows the chloroplastic fatty acid synthesis genes related to SA and JA. We also removed the two GO analyses, FigS5 and FigS6.

Round 2

Reviewer 1 Report

Comments and Suggestions for Authors

The authors have addressed the comments and made significant improvements to the manuscript. It can now be accepted pending a final check of grammar and English.

Comments on the Quality of English Language

NA

Author Response

Comments 1: The authors have addressed the comments and made significant improvements to the manuscript. It can now be accepted pending a final check of grammar and English.

Response 1Thank you again for spending time on the manuscript.

Reviewer 2 Report

Comments and Suggestions for Authors

The question for my answers has been unsatisfactory and now I have more doubts on the validity of the results.

a) Why authors are using albinic protoplast for the bifc? which is the point if they want to see an interaction in the protoplast.

b) Authors say they have three independent overexpression lines. But only show results from one. Why? It is difficult to evaluate if the results is due to the gene overexpressed or a position effect.

c) Again, please describe which kind of vesicles are stained by wortmannin; early endosome? late endosome/prevacuolar compartment? vesicles from the golgi? 

Author Response

Comments 1: Why authors are using albinic protoplast for the bifc? which is the point if they want to see an interaction in the protoplast.

Response 1: We used albinic protoplast for the following considerations: (1) Pik1-H4 and Pik2-H4 are located in the cytoplasm and nucleus, we did not expect the Pik-H4 pair to appear in chloroplasts at first. (2) Chloroplast-less protoplasts are useful for visualizing the cytoplasm and nucleus location. (3) We also wondered whether the Pik-H4 protein pair interacts with OsRRM2 in cells without chloroplasts. And if so, where does the interaction take place? We supplemented the statement and discussion in lines 145-146, 283-286.

Comments 2: Authors say they have three independent overexpression lines. But only show results from one. Why? It is difficult to evaluate if the results is due to the gene overexpressed or a position effect.

Response 2: We are sorry for misinterpreting your last corresponding comment. We chose 3 different lines of overexpression or knockout plants (Supplementary Figure 3), tested for inoculation, and sampled 10 lesions to measure the lesion length and spore ratio at the lesion areas from 3 different lines. We altered Figure 3a to show 3 different lines, Figure 3b and c to show 10 individual data in each column, and the description in lines 173-176.

Comments 3: Again, please describe which kind of vesicles are stained by wortmannin; early endosome? late endosome/prevacuolar compartment? vesicles from the golgi? 

Response 3: Agree. We did not realize the importance of certain types of vesicles. ARA6 is found in the membrane, endoplasmic reticulum, endosome membrane, and early endosome membrane (DOI: 10.1038/ncb2270, 10.1104/pp.109.142349). When treated with wortmannin, ARA6 is found in endosomal vesicles only (DOI: 10.1038/ncb2270). In these two studies, ARA6 is described as ‘in endosomal organelles’ or ‘endosomal membrane’, thus, the type of endosomal vesicles is not clear. The corresponding description ‘endosomal vesicle’ is added in lines 138-139, 143-144.

Reviewer 4 Report

Comments and Suggestions for Authors

No further comments. Reforms in the presentation of results have improved the presentation quality and clarity and relevant justifications are in certain cases supportive, and in other cases adequate.

I accept the overall conclusions. I think the authors are just/fair with what is found. Though in figure 4 the p-values in many cases are a point of consideration I think that in the OE plants the fold changes can be accepted as a trend (considering that in ko plants no substantial differences can be seen (fold & p-values combination)).

Author Response

Comments 1: No further comments. Reforms in the presentation of results have improved the presentation quality and clarity and relevant justifications are in certain cases supportive, and in other cases adequate.

I accept the overall conclusions. I think the authors are just/fair with what is found. Though in figure 4 the p-values in many cases are a point of consideration I think that in the OE plants the fold changes can be accepted as a trend (considering that in ko plants no substantial differences can be seen (fold & p-values combination)).

Response 1Agree. We modified lines 204-205 with a more specific description.